# Serum GDF-15 Levels in Patients with Parkinson's Disease, Progressive Supranuclear Palsy, and Multiple System Atrophy

**Noriyuki Miyaue [1,2,*]** , **Hayato Yabe [2] and Masahiro Nagai [1]**

1 Department of Clinical Pharmacology and Therapeutics, Graduate School of Medicine, Ehime University, Tohon 791-0295, Ehime, Japan; mnagai@m.ehime-u.ac.jp
2 Department of Neurology, Saiseikai Matsuyama Hospital, Matsuyama 791-8026, Ehime, Japan; yabehayato@matsuyama.saiseikai.or.jp
* Correspondence: miyaue@m.ehime-u.ac.jp; Tel.: +81-89-960-5095

**Abstract:** Serum growth differentiation factor 15 (GDF-15) levels are elevated in patients with Parkinson's disease (PD) and may help differentiate these patients from healthy individuals. We aimed to clarify whether serum GDF-15 levels can help differentiate PD from atypical parkinsonian syndromes and determine the association between serum GDF-15 levels and clinical parameters. We prospectively enrolled 46, 15, and 12 patients with PD, progressive supranuclear palsy (PSP), and multiple system atrophy (MSA), respectively. The serum GDF-15 level in patients with PD (1394.67 ± 558.46 pg/mL) did not differ significantly from that in patients with PSP (1491.27 ± 620.78 pg/mL; $p = 0.573$) but was significantly higher than that in patients with MSA (978.42 ± 334.66 pg/mL; $p = 0.017$). Serum GDF-15 levels were positively correlated with age in patients with PD ($r = 0.458$; $p = 0.001$); PSP ($r = 0.565$; $p = 0.028$); and MSA ($r = 0.708$; $p = 0.010$). After accounting for age differences, serum GDF-15 levels did not differ significantly between patients with PD and MSA ($p = 0.114$). Thus, age has a strong influence on serum GDF-15 levels, which may not differ significantly between patients with PD and atypical parkinsonian syndromes such as PSP and MSA.

**Keywords:** GDF-15; Parkinson's disease; multiple system atrophy; progressive supranuclear palsy; parkinsonian syndrome

## 1. Introduction

Parkinson's disease (PD) is the second most common neurodegenerative disorder and is clinically characterized by tremor, rigidity, bradykinesia, postural instability, as well as a variety of non-motor symptoms. In addition to PD, other neurodegenerative diseases also coincide with parkinsonism, including progressive supranuclear palsy (PSP) and multiple system atrophy (MSA), and are referred to as atypical parkinsonian syndromes. PSP is characterized by parkinsonism, vertical supranuclear palsy, and early postural instability and falls; while MSA is characterized by parkinsonism and cerebellar dysfunction associated with autonomic dysfunction and is classified into MSA with predominant parkinsonism (MSA-P) or MSA with predominant cerebellar ataxia (MSA-C), according to the predominant symptoms. Distinguishing these atypical parkinsonian syndromes from PD is important because the functional decline caused by disease progression in these syndromes is usually more rapid than that in PD [1]. However, the differential diagnosis between PD and these atypical parkinsonian syndromes is often difficult, especially in the early stages of onset when symptoms other than parkinsonism are not evident, and no established differential biomarkers currently exist.

Growth differentiation factor 15 (GDF-15), which is identical to macrophage inhibitory cytokine-1, is a member of the transforming growth factor-β superfamily and can be induced in response to cellular stress [2]. GDF-15 is expressed in a broad range of tissues,

including the central and peripheral nervous systems [3]. A variety of diseases have been reported to be associated with elevated GDF-15 blood levels, including chronic heart failure, coronary artery disease, chronic liver disease, chronic kidney disease, diabetes mellitus (DM), and tumor-induced anorexia [4–9]. We recently reported that patients with PD show significantly higher serum levels of GDF-15 than the age-matched healthy controls [10]. However, few studies have reported GDF-15 levels in atypical parkinsonian syndromes other than PD [11,12].

In this study, serum GDF-15 levels were measured in patients with PD, as well as in patients with PSP and MSA, to investigate the potential usefulness of serum GDF-15 levels as a biomarker in differential diagnosis. We also examined the clinical parameters associated with serum GDF-15 levels in each patient group.

## 2. Materials and Methods

### 2.1. Study Design and Participants

We prospectively enrolled patients diagnosed with PD, PSP, or MSA according to the Movement Disorder Society Criteria [13–15] from November 2020 to April 2023. The exclusion criteria were as follows: (i) patients diagnosed with or treated for malignancy in the past year; (ii) patients undergoing treatment for myocardial infarction or heart failure; (iii) patients with chronic hepatitis or liver cirrhosis; or (iv) patients with serum creatinine levels higher than 1.5 mg/dL. In addition to obtaining blood samples for measurement of serum GDF-15 levels, we collected data from all patients, including age, sex, disease duration, modified Rankin scale (mRS) score, and presence of comorbid DM.

This study was approved by the Institutional Review Board for Clinical Research Ethics of Ehime University (Approval code: 1906007; Approval date: 26 September 2022) and Saiseikai Matsuyama Hospital (Approval code: S20-07; Approval date: 5 April 2021) and was conducted in accordance with the Declaration of Helsinki. Written informed consent was obtained from all participants.

### 2.2. Determination of Serum GDF-15 Levels

The blood samples were centrifuged at $1000 \times g$ for 10 min, and serum was collected and stored at $-80\ ^\circ$C for analysis in our laboratory at Ehime University. Serum GDF-15 levels were measured in duplicate from 50 µL of serum using an enzyme-linked immunosorbent assay kit (R&D Systems, Minneapolis, MN, USA) in accordance to the manufacturer's instructions. Test samples were taken once from the same patients.

### 2.3. Statistical Analysis

Continuous data were expressed as mean $\pm$ standard deviation. Fisher's exact test was used to compare categorical variables between groups. The *t*-test or ANOVA was used to compare continuous variables. Spearman's rank correlation test was used to identify correlations between serum GDF-15 levels and other parameters. Statistical significance was defined as a two-tailed *p*-value of <0.05. All analyses were conducted using R statistical software (version 4.3.0, The R Project for Statistical Computing, Vienna, Austria).

## 3. Results

The study population included 46 patients with PD, 15 patients with PSP, and 12 patients with MSA. The clinical profiles of these patients are shown in Table 1. The mean age was lower in patients with MSA (67.00 $\pm$ 8.50 years) than in patients with PD (73.00 $\pm$ 8.25 years) and PSP (72.00 $\pm$ 8.20 years) (*p* = 0.089). Patients with PD had a significantly longer disease duration (8.24 $\pm$ 6.35 years; *p* = 0.007) and higher mRS (2.30 $\pm$ 0.94; *p* < 0.001). Serum GDF-15 levels were 1394.67 $\pm$ 558.46 pg/mL in patients with PD; 1491.27 $\pm$ 620.78 pg/mL in patients with PSP; and 978.42 $\pm$ 334.66 pg/mL in patients with MSA (Figure 1). Patients with PD showed significantly higher serum GDF-15 levels than the patients with MSA (*p* = 0.017) while there was no significant difference between patients with PD and those with PSP (*p* = 0.573). Incidentally, in our previous study, serum GDF15 levels were

1472.22 ± 820.12 pg/mL in patients with PD (72.44 ± 8.84 years) and 1092.83 ± 543.97 pg/mL in healthy individuals (71.93 ± 8.86 years) [10].

**Table 1.** Clinical characteristics of patients with PD, PSP, and MSA.

|  | **PD** | **PSP** | **MSA** | *p* |
|---|---|---|---|---|
| *N* | 46 | 15 | 12 | |
| Age, years | 73.00 ± 8.25 | 72.00 ± 8.20 | 67.00 ± 8.50 | 0.089 |
| Males, *n* (%) | 24 (52.2) | 8 (53.3) | 9 (75.0) | 0.354 |
| Disease duration, years | 8.24 ± 6.35 | 4.00 ± 2.39 | 4.17 ± 2.12 | 0.007 |
| mRS score | 2.30 ± 0.94 | 3.73 ± 0.70 | 3.08 ± 1.00 | <0.001 |
| Comorbid DM, *n* (%) | 5 (10.9) | 5 (33.3) | 3 (25.0) | 0.110 |

Values are shown as mean ± standard deviation or *n* (%). Differences between groups were analyzed with ANOVA or Fisher's exact test. DM, diabetes mellitus; GDF-15, growth differentiation factor 15; mRS, modified Rankin scale; MSA, multiple system atrophy; PD, Parkinson's disease; PSP, progressive supranuclear palsy.

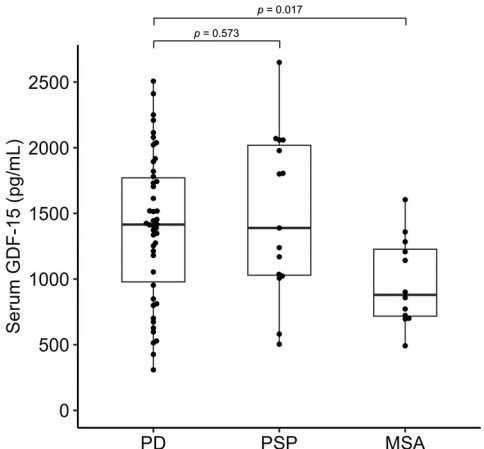

**Figure 1.** Serum GDF-15 levels in patients with PD, PSP, and MSA. The box-and-whiskers plot shows the median (line) and the lower and upper interquartile range values. The *t*-test was used for comparison with the PD group. GDF-15, growth differentiation factor 15; MSA, multiple system atrophy; PD, Parkinson's disease; PSP, progressive supranuclear palsy.

Next, we examined the association between serum GDF-15 levels and clinical parameters in each patient group. In patients with PD, serum GDF-15 levels showed a significant positive correlation with age (*r* = 0.458; *p* = 0.001); disease duration (*r* = 0.314; *p* = 0.034); and mRS score (*r* = 0.407; *p* = 0.005; Figure 2A–C). Serum GDF-15 levels did not differ significantly between female (*n* = 22, 1314.91 ± 526.80 pg/mL) and male patients (*n* = 24, 1467.79 ± 587.49 pg/mL; *p* = 0.359; Figure 2D). Patients with comorbid DM (*n* = 5, 1784.00 ± 337.62 pg/mL) tended to have higher serum GDF-15 levels than those without comorbid DM (*n* = 41, 1347.20 ± 564.09 pg/mL; *p* = 0.099; Figure 2E).

In patients with PSP, serum GDF-15 levels showed a significant positive correlation with age (*r* = 0.565; *p* = 0.028) but not with disease duration (*r* = −0.319; *p* = 0.247) or mRS score (*r* = 0.130; *p* = 0.643; Figure 3A–C). No significant difference was observed in serum GDF-15 levels between female (*n* = 7, 1732.29 ± 384.36 pg/mL) and male patients (*n* = 8, 1280.38 ± 731.57 pg/mL; *p* = 0.167, Figure 3D). Patients with comorbid DM (*n* = 5, 1940.60 ± 463.60 pg/mL) tended to have higher serum GDF-15 levels than those without comorbid DM (*n* = 10, 1266.60 ± 579.39 pg/mL; *p* = 0.052, Figure 3E).

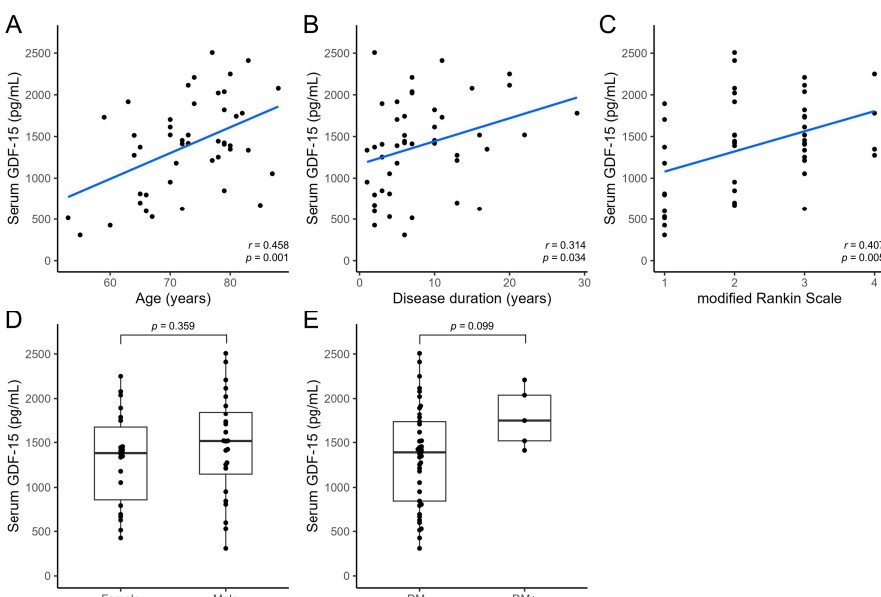

**Figure 2.** Comparison of serum GDF-15 levels with clinical parameters in patients with Parkinson's disease using the *t*-test or Spearman's rank correlation test. Scatter plots show the association of serum GDF-15 levels with age (**A**), disease duration (**B**), and modified Rankin scale score (**C**). Box-and-whiskers plots show the association of serum GDF-15 levels with sex (**D**), and the presence of comorbid DM (**E**) with the median (line) and lower and upper interquartile range values. DM, diabetes mellitus; GDF-15, growth differentiation factor 15.

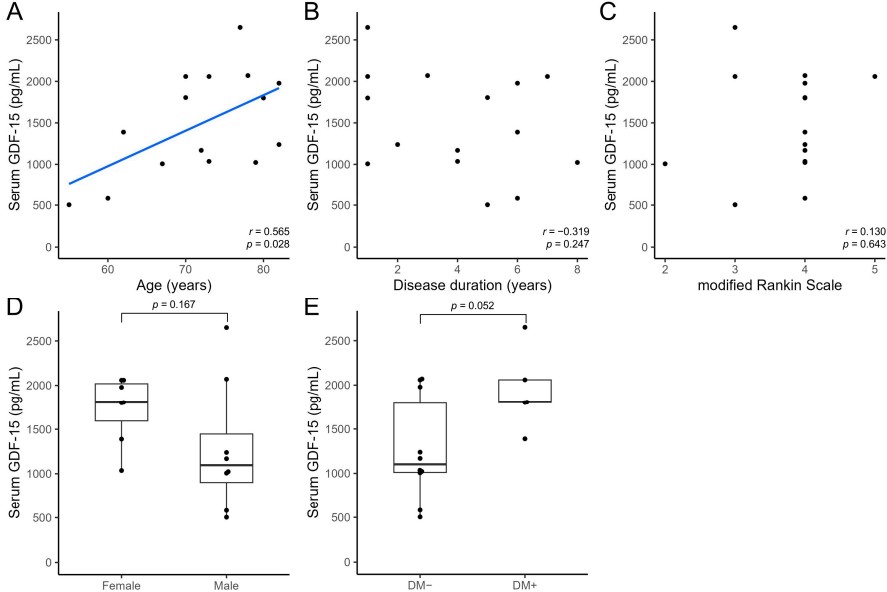

**Figure 3.** Comparison of serum GDF-15 levels with clinical parameters in patients with progressive supranuclear palsy using the *t*-test or Spearman's rank correlation test. Scatter plots show the association of serum GDF-15 levels with age (**A**), disease duration (**B**), and modified Rankin scale score (**C**). Box-and-whiskers plots show the association of serum GDF-15 levels with sex (**D**) and presence of comorbid DM (**E**) with the median (line) and lower and upper interquartile range values. DM, diabetes mellitus; GDF-15, growth differentiation factor 15.

In patients with MSA, serum GDF-15 levels showed a significant positive correlation with age ($r = 0.708$; $p = 0.010$) and a significant negative correlation with disease duration ($r = -0.638$; $p = 0.026$), but not with the mRS score ($r = -0.287$; $p = 0.365$; Figure 4A–C). Serum GDF-15 levels showed no significant difference between female

($n$ = 3, 1039.33 ± 277.67 pg/mL) and male patients ($n$ = 9, 958.11 ± 364.50 pg/mL; $p$ = 0.734; Figure 4D), nor between patients with comorbid DM ($n$ = 3, 1015.33 ± 278.51 pg/mL) and those without comorbid DM ($n$ = 9, 966.11 ± 365.95 pg/mL; $p$ = 0.837; Figure 4E). Regarding clinical subtypes, no significant difference was observed in serum GDF-15 levels between MSA-C ($n$ = 7, 946.57 ± 330.33 pg/mL) and MSA-P ($n$ = 5, 1023.00 ± 374.23 pg/mL; $p$ = 0.716; Figure 4F).

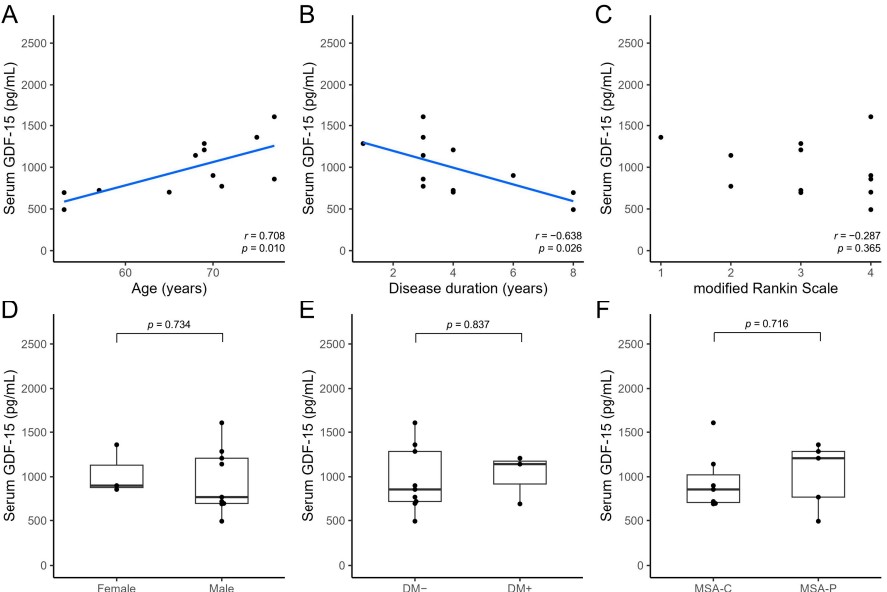

**Figure 4.** Comparison of serum GDF-15 levels with clinical parameters in patients with multiple system atrophy using the *t*-test or Spearman's rank correlation test. Scatter plots show the association of serum GDF-15 levels with age (**A**), disease duration (**B**), and modified Rankin scale score (**C**). Box-and-whiskers plots show the association of serum GDF-15 levels with sex (**D**), presence of comorbid DM (**E**), and clinical subtypes (MSA-C/MSA-P) (**F**) with the median (line) and lower and upper interquartile range values. DM, diabetes mellitus; GDF-15, growth differentiation factor 15; MSA-C, multiple system atrophy with predominant cerebellar ataxia; MSA-P, multiple system atrophy with predominant parkinsonism.

As shown above, serum GDF-15 levels showed a positive correlation with age in all patient groups. A logistic regression analysis with age and serum GDF-15 levels as independent variables and disease group as a dependent variable showed that the difference in serum GDF-15 levels between PD and MSA patients was not significant ($p$ = 0.114).

## 4. Discussion

This study showed that the serum GDF-15 levels in patients with PSP were comparable to those in patients with PD. This is the first study to report GDF-15 levels in patients with PSP. On the other hand, patients with MSA had lower serum GDF-15 levels than the patients with PD, but this finding was not significant when adjusted for age. A recent study reported that the serum GDF-15 levels in patients with MSA are greater than those in patients with PD [12], which is different from our results. Another study on GDF-15 levels in atypical parkinsonian syndromes showed that cerebrospinal fluid GDF-15 levels in patients with Lewy body dementia were significantly higher than those in patients with PD and dementia but lower than those in patients with PD without dementia [11]. The findings of this study and these previous studies indicate that the clinical differentiation of PD from atypical parkinsonian syndromes, solely based on GDF-15 levels, may be difficult.

In the clinical characteristics of the patients in this study, patients with PD had a significantly longer disease duration. This is consistent with the epidemiological feature that patients with PD generally have a longer disease duration than the patients with PSP and MSA [16].

The patients with PD in this study showed a significantly positive correlation of serum GDF-15 levels with disease duration and mRS score. Previous studies have also reported a positive correlation between serum GDF-15 levels, disease duration, cerebrospinal fluid GDF-15 levels, and Hoehn and Yahr scale scores in patients with PD [11,17]. These findings suggest that GDF-15 levels may reflect disease severity in patients with PD.

Among the patients with PSP in this study, serum GDF-15 levels were higher in female patients than in male patients although the ages of male patients (71.25 ± 9.74 years) and female patients (72.86 ± 6.69 years) were similar ($p = 0.772$). On the other hand, female patients (57.1%) showed a higher rate of comorbid DM than male patients (12.5%). Since patients with PD who have comorbid DM tended to have higher serum GDF-15 levels in this study, and patients with DM had higher blood GDF-15 levels than the healthy participants [18], the difference in serum GDF-15 levels distinguished by sex in patients with PSP may have been influenced by the differences in the rate of comorbid DM.

The patients with MSA in this study showed no difference in serum GDF-15 levels in relation to disease type (MSA-C/MSA-P), similar to a previous report [12]. However, in contrast to that report, this study showed a negative correlation between serum GDF-15 levels and disease duration in patients with MSA. The inverse association between serum GDF-15 concentration and disease duration was probably due to the small number of patients, and the bias toward younger patients in the group, with longer disease duration in this study.

This study showed that the serum GDF-15 level is a common factor showing a significant positive correlation with age in patients with PD, PSP, and MSA. GDF-15 is the plasma protein most strongly associated with age, and its levels increase linearly with increasing age in healthy participants [19]. In addition, serum GDF-15 levels significantly correlate with aging biomarkers, including telomere length, telomerase activity, and the expression of human telomerase reverse transcriptase [20], with GDF-15 also being highly expressed in aging brains [21]. Several studies, including our previous report, have shown that the serum GDF-15 levels increase significantly with age in patients with PD [10,17]. This study suggests an association between aging and serum GDF-15 levels in patients with atypical parkinsonian syndromes, such as PSP and MSA as well as in PD.

A major limitation of this study is that the small number of patients included in this study, especially those with PSP and MSA, may not accurately reflect real world situations. Furthermore, serum GDF-15 levels in atypical parkinsonian syndromes other than PSP and MSA, such as corticobasal syndrome and Lewy body dementia, were not examined. The difficulty in distinguishing PD from atypical parkinsonian syndromes based on serum GDF-15 levels alone will need to be confirmed by increasing the number of target diseases and patients. Another limitation is that this study measured serum GDF-15 levels in all patients at only one time point. Serum GDF-15 levels are associated with all-cause mortality [22,23], and elevated plasma GDF-15 levels can predict the development of anemia in the elderly [24]. Longitudinal examinations are needed to determine whether the serum GDF-15 level is associated with prognosis in patients with PD and atypical parkinsonian syndromes.

## 5. Conclusions

In conclusion, this study showed that the serum GDF-15 levels did not differ significantly between patients with PD and those with PSP or MSA. Furthermore, serum GDF-15 levels showed a significant positive correlation with age in all patient groups (PD, PSP, and MSA). Further studies are required to confirm the utility of serum GDF-15, as a diagnostic and prognostic biomarker, in patients with PD and atypical parkinsonian syndromes.

**Author Contributions:** N.M. conceived and designed the study; N.M. and H.Y. acquired the data; N.M. analyzed the data and drafted the manuscript; H.Y. and M.N. critically revised the manuscript; All authors have read and agreed to the published version of the manuscript.

**Funding:** This research was funded by JSPS Kakenhi (Grant Number JP21K15699).

**Institutional Review Board Statement:** This study was approved by the Institutional Review Board for Clinical Research Ethics of Ehime University (Approval code: 1906007; Approval date: 26 September 2022) and Saiseikai Matsuyama Hospital (Approval code: S20-07; Approval date: 5 April 2021).

**Informed Consent Statement:** Informed consent was obtained from all patients involved in this study.

**Data Availability Statement:** The data supporting the findings of this study are available on reasonable request from the corresponding author.

**Conflicts of Interest:** The authors declare no conflict of interest.

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
