# Peer review of "Serum GDF-15 Levels in Patients with Parkinson’s Disease, Progressive Supranuclear Palsy, and Multiple System Atrophy"

_2035-8377, doi:10.3390/neurolint15030066_

Round 1

Reviewer 1 Report

This clinical study has tested serum growth differentiation factor 15 (GDF-15) levels in patients with Parkinson's disease, progressive supranuclear palsy, and multiple system atrophy. The authors hypothesised that GDF-15 could be helpful as a biomarker for earlier diagnosis of named disorders, but the data do not directly support this statement. There are additional concerns throughout the sections of the paper that require attention.

Methods:

Please specify the volume of the blood sample used for analysis.

Were the samples taken from the patients after fasting or not?

Was the value of GDF-15 levels from duplicate measurements averaged before (line 74)?

Specify that there were no replicates - test samples were taken once from the same patients.

What was the reason for applying the Mann-Whitney U or the Kruskal-Wallis tests? These tests are nonparametric, but the authors show the data as mean with a standard deviation that assumes the data are normally distributed. Did the authors check the data for normality? If the data are normally distributed, then the ANOVA test should be used for multiple group comparisons.

Having a control (healthy) group is necessary (or at least referring to the data from the authors’ previous studies).

 Results:

Add the p values on the plots to see if there is a significant difference between experimental groups.

How did the authors adjust serum GDF-15 levels for age (line 169)? Please explain and add the data.

The text would benefit from improvements in the English language.

Author Response

Response to Reviewer 1 Comments

Point 1: Please specify the volume of the blood sample used for analysis.

Response 1: We would like to thank the reviewer for evaluating our manuscript and for the comment. Serum GDF-15 levels were measured in duplicate from 50 μL of serum using an enzyme-linked immunosorbent assay kit (R&D Systems, Minneapolis, MN, USA) according to the manufacturer’s instructions. We have added the following sentence to the revised manuscript:
“Serum GDF-15 levels were measured in duplicate from 50 μL of serum using an enzyme-linked immunosorbent assay kit (R&D Systems, Minneapolis, MN, USA) according to the manufacturer’s instructions.”  (Lines 73 to 76)

Point 2: Were the samples taken from the patients after fasting or not?

Response 2: We would like to thank the reviewer for the question. In this study, no restrictions were placed on the timing of blood collection.

Point 3: Was the value of GDF-15 levels from duplicate measurements averaged before (line 74)?

Response 3: We would like to thank the reviewer for the question. We averaged the results of the duplicate measurements.

Point 4: Specify that there were no replicates - test samples were taken once from the same patients.

Response 4: We would like to thank the reviewer for the comment. We have added the following sentence to the revised manuscript:
“Test samples were taken once from the same patients.” (Line 76)

Point 5: What was the reason for applying the Mann-Whitney U or the Kruskal-Wallis tests? These tests are nonparametric, but the authors show the data as mean with a standard deviation that assumes the data are normally distributed. Did the authors check the data for normality? If the data are normally distributed, then the ANOVA test should be used for multiple group comparisons.

Response 5: We would like to thank the reviewer for the comment. Since the data were normally distributed, t-test and ANOVA were used for between-group comparisons. We have revised the statistical results in the revised manuscript.

Point 6: Having a control (healthy) group is necessary (or at least referring to the data from the authors’ previous studies).

Response 6: We would like to thank the reviewer for the comment. We have added the following sentence to the revised manuscript:
“Incidentally, in our previous study, serum GDF15 levels were 1472.22 ± 820.12 pg/mL in patients with PD (72.44 ± 8.84 years) and 1092.83 ± 543.97 pg/mL in healthy individuals (71.93 ± 8.86 years).” (Lines 97 to 99)

Point 7: Add the p values on the plots to see if there is a significant difference between experimental groups.

Response 7: We would like to thank the reviewer for the comment. We have added the p values on the plots in the revised manuscript.

Point 8: How did the authors adjust serum GDF-15 levels for age (line 169)? Please explain and add the data.

Response 8: We would like to thank the reviewer for the comment. We have added the following sentence to the revised manuscript:
“A logistic regression analysis with age and serum GDF-15 levels as independent variables and disease group as a dependent variable showed that the difference in serum GDF-15 levels between PD and MSA patients was not significant (p = 0.114).” (Lines 168 to 170)

Point 9:

The text would benefit from improvements in the English language.

Response 9: We would like to thank the reviewer for the comment. We had a native English speaker proofread this paper. We would like to thank Editage (www.editage.jp) for English language editing.

Reviewer 2 Report

Growth differentiation factor 15 (GDF-15) belongs to the transforming growth factor-β family and acts as an inflammatory marker. It exhibits increased levels in various diseases, including tumors, ischemic heart diseases, metabolic disorders, and neurodegenerative disorders such as Parkinson's disease (PD). GDF-15 shows potential for distinguishing patients with PD from healthy individuals. This manuscript aims to assess the utility of serum GDF-15 levels in differentiating patients with PD from those with atypical parkinsonism, specifically progressive supranuclear palsy (PSP) and multiple system atrophy (MSA).

Comments and Suggestions:

1            In Line 233, it's essential for the authors to provide the approval number from their institute's review board. (Major). 

2            Given that serum GDF-15 levels can rise in various conditions such as tumors, diabetes, and ischemic heart disease, it is crucial to address potential confounding factors in Table 1 or within the Materials and Methods section. Furthermore, since the PD group exhibited significantly longer disease durations compared to the other groups, the potential implications of this difference should be discussed (Major). 

3            Although the authors have noted that the numbers of PSP and MSA patients were lower than those with PD (Line 208), the study's total sample size remains quite small: 46 PD patients, 15 PSP patients, and 12 MSA patients. As a result, the study's results may not accurately reflect real-world situations, regardless of significance. This limitation should be explicitly stated (Major).

4            Given the limited patient numbers (Lines 114-130), to demonstrate the impact of diabetes on GDF-15 may be challenging. As a result, it might be prudent to omit this portion from the manuscript, including the Results and Discussion sections (Minor).

5            To enhance clarity, consider incorporating the statistical methods used in the captions of tables and the legends of figures. This will aid readers in understanding the statistical analyses conducted (Major).

6            In figures 2, 3, and 5, adding regression lines and p-values as figure notes would enhance readability and facilitate interpretation. This is a major suggestion (Major).

In conclusion, this study did not yield novel or significant insights into the potential of GDF-15 as a diagnostic tool for distinguishing PD from related diseases (PSP and MSA).

Author Response

Response to Reviewer 2 Comments

Point 1: In Line 233, it's essential for the authors to provide the approval number from their institute's review board. (Major).

Response 1: We would like to thank the reviewer for evaluating our manuscript and for the comment. We have added the following sentence to the revised manuscript:
“This study was approved by the institutional review board for clinical research ethics of Ehime University (1906007) and Saiseikai Matsuyama Hospital (S20-07).” (Lines 243 to 245)

Point 2: Given that serum GDF-15 levels can rise in various conditions such as tumors, diabetes, and ischemic heart disease, it is crucial to address potential confounding factors in Table 1 or within the Materials and Methods section. Furthermore, since the PD group exhibited significantly longer disease durations compared to the other groups, the potential implications of this difference should be discussed (Major).

Response 2: We would like to thank the reviewer for the comment. The study addressed confounding factors to the extent possible by excluding patients diagnosed with or treated for malignancy in the past year, patients undergoing treatment for myocardial infarction or heart failure, patients with chronic hepatitis or liver cirrhosis, and patients with serum creatinine levels higher than 1.5 mg/dL. We have added the following discussion regarding the PD group having a significantly longer disease duration than the other groups to the revised manuscript:

In the clinical characteristics of the patients in this study, patients with PD had a significantly longer disease duration. This is consistent with the epidemiological feature that patients with PD generally have a longer disease duration than patients with PSP and MSA [16].” (Lines 184 to 187)

Point 3 Although the authors have noted that the numbers of PSP and MSA patients were lower than those with PD (Line 208), the study's total sample size remains quite small: 46 PD patients, 15 PSP patients, and 12 MSA patients. As a result, the study's results may not accurately reflect real-world situations, regardless of significance. This limitation should be explicitly stated (Major).

Response 3: We would like to thank the reviewer for the comment. We have added the following sentence to the revised manuscript:
“A major limitation of this study is that the small number of patients included in the study, especially those with PSP and MSA, may not accurately reflect real-world situations.” (Lines 219 to 221)

Point 4: Given the limited patient numbers (Lines 114-130), to demonstrate the impact of diabetes on GDF-15 may be challenging. As a result, it might be prudent to omit this portion from the manuscript, including the Results and Discussion sections (Minor).

Response 4: We would like to thank the reviewer for the comment. As mentioned above, we have attempted to exclude confounding factors as much as possible in this study, and we hope to mention that DM can also be a contributing factor affecting serum GDF15 levels.

Point 5: To enhance clarity, consider incorporating the statistical methods used in the captions of tables and the legends of figures. This will aid readers in understanding the statistical analyses conducted (Major).

Response 5: We would like to thank the reviewer for the comment. We have added the statistical methods in the captions of tables and the legends of figures in the revised manuscript.

Point 6: In figures 2, 3, and 5, adding regression lines and p-values as figure notes would enhance readability and facilitate interpretation. This is a major suggestion (Major).

Response 6: We would like to thank the reviewer for the comment. We have added the regression coefficient and p values on the plots in the revised manuscript.

Round 2

Reviewer 1 Report

The authors have addressed the concerns that were brought up during the paper's review.

Minor English language editing is required, especially for the added text.

Reviewer 2 Report

No more comment